# Annual global grided livestock mapping from 1961 to 2021

Zhenrong Du<sup>1</sup>, Le Yu<sup>2,3,4</sup>, Yue Zhao<sup>2</sup>, Xinyue Li<sup>1</sup>, Xiaoxuan Liu<sup>5</sup>, Xiyu Li<sup>2</sup>, Pengyu Hao<sup>6</sup>, Zhongxin Chen<sup>6</sup>, Xiaorui Ma<sup>1</sup>, and Hongyu Wang<sup>1</sup>

Correspondence to: Le Yu (leyu@tsinghua.edu.cn)

Abstract. Understanding global livestock dynamics is essential for global food security, public health, socio-economic and sustainable development. This study developed an automated global livestock mapping framework that integrated Food and Agriculture Organization Corporate Statistical Database (FAOSTAT) and the Random Forest regression model. By implementing the mapping scheme on Google Earth Engine (GEE), we develop the first annual gridded livestock of the world (AGLW), covering the period from 1961 to 2021 at a spatial resolution of 5 km. The annual maps of AGLW were then evaluated from three perspectives: model level, finer-scale statistic level, and pixel level, with correlation coefficients (*r*) of 0.54-0.73, 0.79-0.98, and 0.73-0.83, respectively. The AGLW maps reveal the spatio-temporal dynamics of global livestocks over the past six decades, highlighting both global expansion and localized fluctuations, such as the notable increase in pig stock in China and the decline in horse stock in Poland. By offering a reliable and continuous dataset, AGLW overcomes the limitations of existing livestock mapping products in terms of spatio-temporal continuity and resolution. This dataset serves as a crucial resource for enhancing our understanding of global livestock dynamics, informing policy decisions, guiding sustainable agricultural practices, and promoting resilience in both ecological and human systems. The full archive of AGLW is available at https://doi.org/10.5281/zenodo.11545701 (Du et al., 2025).

#### 15 1 Introduction

Livestock encompassing various animal species raised for economic, agricultural, and cultural purposes, play a pivotal role in global food security, livelihoods, and socio-economic development (Godfray et al., 2018; Tilman and Clark, 2014; Steinfeld et al., 2006). The contribution of livestock to human nutrition, income generation, and rural employment underscores its significance in sustaining livelihoods worldwide (Bonilla-Cedrez et al., 2023; Meisner et al., 2022; Rahimi et al., 2022). With the continuous growth of the global population over the past decades, the demand for livestock has surged, resulting in a rapid

<sup>&</sup>lt;sup>1</sup>School of Information and Communication Engineering, Dalian University of Technology, Dalian 116024, China

<sup>&</sup>lt;sup>2</sup>Department of Earth System Science, Ministry of Education Key Laboratory for Earth System Modeling, Institute for Global Change Studies, Tsinghua University, Beijing 100084, China

<sup>&</sup>lt;sup>3</sup>Ministry of Education Ecological Field Station for East Asian Migratory Birds, Department of Earth System Science, Tsinghua University, Beijing 100084, China

<sup>&</sup>lt;sup>4</sup>Tsinghua University (Department of Earth System Science)- Xi'an Institute of Surveying and Mapping Joint Research Center for Next-Generation Smart Mapping, Beijing 100084, China

<sup>&</sup>lt;sup>5</sup>Aerospace Information Research Institute, Chinese Academy of Sciences, Beijing 100190, China

<sup>&</sup>lt;sup>6</sup>Food and Agriculture Organization of the United Nations, Viale delle Terme di Caracalla, 00153 Rome, Italy

increase in their stocks (Perry et al., 2013; Bouwman et al., 2013; McMichael et al., 2007). For instance, the stock of goat has risen from  $3.49 \times 10^8$  heads in the 1960s to  $1.12 \times 10^9$  heads in the 2020s (approximately 2-3 times), while the chicken stock has increased from  $3.91 \times 10^9$  in the 1960s to  $2.60 \times 10^10$  in the 2020s (approximately 5-6 times) (FAO, 2024). Moreover, livestock constitute slightly over 1/9 of all vertebrate biomass, with estimates suggesting that the combined mass of Earth's livestock, around 100 million metric tons (approximately 110 million tons), exceeds that of human beings, wild birds, and wild mammals combined (Britannica, 2024). Furthermore, serving as the primary source of animal protein (such as milk, meat, and eggs), as well as providing fertilizer for crop production, livestocks contribute significantly to the livelihoods and nutrition of impoverished households in low- and middle-income countries (Godfray et al., 2018; Van Boeckel et al., 2019; Baltenweck et al., 2024). Additionally, they are closely linked to the spread of food-borne diseases and the emergence and transmission of zoonotic diseases (Slingenbergh, 2013; Gilbert et al., 2017; Rulli et al., 2021).

Therefore, from the perspective of economic and social development, public health, carbon emissions, ecological and environmental protection, changes in the spatial distribution of livestock are of significant importance (Strassburg et al., 2020; Li et al., 2023; Herrera et al., 2017). Reliable and continuous global livestock maps are crucial for assessing the impacts of livestock farming, tracing epidemiological patterns, analyzing spatiotemporal changes, and informing planning and policy-making to promote a safe, sustainable, and equitable livestock industry (Bonilla-Cedrez et al., 2023; Brandt et al., 2018; Shahrabi-Farahani et al., 2024). Currently, the only globally comprehensive and influential livestock mapping datasets are the Gridded Livestock of the World (GLW) series (GLW1, GLW2, GLW3, GLW4) (Wint, 2007; Robinson et al., 2014; Gilbert et al., 2018), which were developed based on the Food and Agriculture Organization Corporate Statistical Database (FAOSTAT) (FAO, 2024). As a dataset covering the entire globe, the GLW series provides detailed livestock spatial distribution data for various years with high resolution (Table 1). In addition to the GLW series, there are several national and sub-national level livestock maps. These include the spatial distribution of livestock in Europe (Neumann et al., 2009), the pig production distribution in China (Zhao et al., 2022), the distribution maps for poultry in China (Prosser et al., 2011), and the annual gridded grazing dataset on the Qinghai–Tibet Plateau from 1982 to 2015 (Meng et al., 2023).

**Table 1.** Gridded Livestock of the World product.

| Product | Year | Spatial resolution                         | Reference               |
|---------|------|--------------------------------------------|-------------------------|
| GLW1    | 2005 | 3 minutes of arc (∼5 km)                   | (Wint, 2007)            |
| GLW2    | 2006 | $0.0083333$ decimal degrees ( $\sim$ 1 km) | (Robinson et al., 2014) |
| GLW3    | 2010 | $0.083333$ decimal degrees ( $\sim 10$ km) | (Gilbert et al., 2018)  |
| GLW4    | 2015 | $0.083333$ decimal degrees ( $\sim 10$ km) | In review               |

GLW: Gridded Livestock of the World

These datasets provide valuable insights into livestock dynamics at different scales, supporting various applications from local to global levels (Clark et al., 2019; Cheng et al., 2023; Halpern et al., 2022). One of the most direct applications is the assessment of regional production, consumption, and trade footprints of animal-source food, which serves as a critical

foundation for regional food supply and industry planning (Thornton et al., 2021; Bonilla-Cedrez et al., 2023; Simões et al., 2021; Rahimi et al., 2022). Additionally, livestock maps are essential for quantifying global and regional ecosystem services and analyzing the impact of overgrazing on grassland ecosystem degradation (Maestre et al., 2022; Meng et al., 2023), and play a crucial role in studying gas emissions, climate change, and global land-use changes caused by agricultural activities (Pendrill et al., 2022; Liu et al., 2022; Williams et al., 2021; Theobald et al., 2020; Rahimi et al., 2021). Furthermore, livestock distribution data has been utilized in public health-related research, such as analyzing the distribution patterns and transmission paths of animal-related infectious diseases, including trypanosomosis, coronavirus, foot-and-mouth disease (Rulli et al., 2021; Baudron and Liégeois, 2020; Michelitsch et al., 2021). However, current research is limited by the absence of temporally continuous and long-term global livestock maps. Consequently, most related applications remain at a single temporal snapshot level, making it challenging to conduct time trajectory analyses that integrate long-term trade records, land cover/use maps, and disease statistics. The shortage of spatial and temporal continuity and resolution in existing livestock maps restricts their further application within this context.

Currently, livestock mapping methods predominantly use annual statistical data from various administrative units, combined with other relevant data such as population, land use/cover, topography, vegetation, climate, etc. Machine learning algorithms are then employed to spatially distribute the statistical data, resulting in maps of different livestock species (Gilbert et al., 2018; Meng et al., 2023). In this study, we used annual FAOSTAT livestock statistics from 1961 to 2021, and employed a Random Forest regression model to construct a long-term annual gridded livestock of the world (AGLW) from 1961 to 2021. We then evaluated the mapping results at the model level, finer-scale statistics level, and pixel level using province/state and county statistical data from other sources, as well as existing livestock mapping products. To our knowledge, the AGLW dataset developed in this study is the first long-term annual global livestock maps. This dataset provides an indispensable data foundation for a broad field of studies, encompassing global land-use change analysis, public health studies, ecosystem monitoring, and sustainable development initiatives within the realm of Earth system science.

### 2 Datasets

### 70 2.1 Mapping datasets

## 2.1.1 Country-level statistics

The foundational data for our annual global livestock mapping was sourced from the Food and Agriculture Organization Corporate Statistical Database (FAOSTAT) (FAO, 2024), which can provide nutrition, food, and agriculture statistics spanning back to 1961 across 245 countries worldwide. FAOSTAT offers a rich time series dataset covering the livestock domain crucial for our analysis. Specifically, "crops and livestock products" dataset within FAOSTAT was used as the primary input. This dataset encompassed annual records dating from 1961 to 2021, which document the numbers of live animals across various species, including buffalo, cattle, goats, horses, sheep, chickens, ducks, and pigs.

# 2.1.2 Mapping features

85

100

Based on the country-level livestock statistics, the theoretical suitable masks were generated using the land cover product, population density, and elevation, which can be obtained from the datasets of FROM-GLC Plus (Yu et al., 2022), World-Pop(WorldPop, 2024), and GTOPO30 (Observation and Center, 2017), respectively (Table 2). Areas with high population density and elevation were firstly masked out from the suitable masks. Considering the feeding differences of grazing livestock (i.e., buffalo, cattle, goats, horses, and sheep) and captive livestock (i.e., chickens, ducks, and pigs), land cover classes of grassland and impervious surface in FROM-GLC Plus were selected for the two different types of livestock to obtain the theoretical suitable masks.

**Table 2.** List of datasets for the generation of theoretical suitable masks.

| Variables          | Datasets      | Years      | Spatial resolution | Source                         |
|--------------------|---------------|------------|--------------------|--------------------------------|
| Land cover         | FROM-GLC Plus | 1982-2022* | 30 m               | (Yu et al., 2022)              |
| Population density | WorldPop      | 2000-2020* | 92.77 m            | (WorldPop, 2024)               |
| Elevation          | GTOPO30       | 1996       | 30 arc seconds     | (Observation and Center, 2017) |

<sup>\*</sup> Annual dataset

In order to spatialize livestock statistics at the country level, environmental and anthropogenic factors that are spatially heterogeneous and affect the spatial distribution of livestock need to be taken into account in the mapping process. Therefore, for the mapping features, we included 12 variables that involved anthropogenic, topography, climate, soil and vegetation according to the previous livestock mapping studies and other possible factors affecting livestock distribution (Gilbert et al., 2018; Meng et al., 2023) (Table 3).

## 2.2 Evaluation Datasets

### 2.2.1 Province/state and county level statistics

The province/state level statistics of livestock numbers were collected from publications provided by statistical offices in different countries. Due to the unavailability of provincial statistical data on a global scale, we selected a typical province/state for each specie of livestock. Specifically, these statistics were collected for cattle in Texas, United States; chicken in California, United States; horses in Kentucky, United States; pigs in Henan, China; buffaloes in Guangxi, China; goats in Inner Mongolia, China; and sheep in New South Wales, Australia. These regions were chosen due to their long-standing history and regional representativeness in livestock farming, along with the availability of comprehensive statistical data. Given the diverse sources of these data, the coverage years vary (Table 4).

The county-level statistics of livestock numbers were gathered and summarized from China Statistical Yearbook (CSY). For statistics in China, the classes of livestock include pigs, dairy cows, beef cattle, other cattle, poultry, sheep and goats. Particularly, these statistics covered the years of 1990, 2002, 2007, 2012, 2017, and provided independent validation dataset

**Table 3.** List of datasets for the mapping features.

| Type           | Variables                                     | Years      | Spatial resolution | Source                         |
|----------------|-----------------------------------------------|------------|--------------------|--------------------------------|
| Anthropogenic  | Population                                    | 2000-2020* | 92.77 m            | (WorldPop, 2024)               |
|                | Distance to cities of 50000 people            | 2015       | 1 km               | (Weiss et al., 2018)           |
| Topography     | Elevation                                     | 1996       | 30 arc seconds     | (Observation and Center, 2017) |
|                | Slope                                         |            |                    |                                |
| Climate & Soil | Precipitation                                 | 1961-2021* | 4638.3 m           | (Abatzoglou et al., 2018)      |
|                | Minimum temperature                           |            |                    |                                |
|                | Maximum temperature                           |            |                    |                                |
|                | Wind-speed at 10m                             |            |                    |                                |
|                | Soil moisture                                 |            |                    |                                |
| Vegetation     | Normalized Difference Vegetation Index (NDVI) | 1981-1999* | 5566 m             | AVHRR                          |
|                |                                               | 2000-2021* | 1 km               | MOD13A2                        |
|                | Green-up and senescence                       | 2001-2021* | 500 m              | MCD12Q2                        |
|                | Number of cycles                              |            |                    |                                |

<sup>\*</sup> Annual dataset

Table 4. Province/state level statistics for typical regions.

| No. | Livestock | Province/state  | Years                | Country       | Source |
|-----|-----------|-----------------|----------------------|---------------|--------|
| 1   | Cattle    | Texas           | 1969/1974/1978/1982/ | United States | USDA   |
| 2   | Chicken   | California      | 1987/1992/1997/2002/ |               |        |
| 3   | Horses    | Kentucky        | 2007/2012/2017/2022  |               |        |
| 4   | Pigs      | Henan           | 1978-2019*           | China         | CSY    |
| 5   | Buffaloes | Guangxi         | 1978-2007*           |               |        |
| 6   | Goats     | Inner Mongolia  | 1985-2021*           |               |        |
| 7   | Sheep     | New South Wales | 1990-2021*           | Australia     | ABS    |
|     |           |                 |                      |               |        |

USDA: United States Department of Agriculture; CSY: China Statistical Yearbook; ABS: Australian Bureau of Statistics

for the livestock global mapping results in China. Considering the differences in species between the two statistics, specie matching was performed. For example, the mapping results of chickens and ducks were combined as poultry, and beef cattle and other cattle of CSY were combined as cattle. These province/state level and county level statistics were then used for the annual evaluation of AGLW by calculating the correlation coefficients (*r*).

<sup>\*</sup>Annual dataset

### 2.2.2 Pixel-level dataset

To further validate the reliability of the AGLW data at the pixel level, we incorporated the GLW2, GLW3, and GLW4 datasets as outlined in Table 1. These datasets were used for pixel-level evaluation for the corresponding years 2006, 2010, and 2015.

We did not use the GLW1 dataset for validation due to its difficult data acquisition and relatively coarse spatial heterogeneity (Robinson et al., 2014). As presented in Table 1, the spatial resolution of the GLW datasets varied from approximately 1 km to 10 km. It's important to note that the range of livestock types covered by these three datasets differs. GLW2 includes only cattle, pigs, and chickens, with a partial distribution map for ducks. In contrast, GLW3 and GLW4 cover the same livestock types as AGLW, including cattle, buffaloes, horses, sheep, goats, pigs, chickens, and ducks.

#### 115 3 Methods

130

The long-term annual livestock mapping procedure developed in this study is presented in Figure 1. Generally, the workflow adheres to the mapping framework outlined in GLW series. FAOSTAT serves as the primary input for country-level statistics and acts as the basis for corrections. To refine these statistics to the city level, we used the GLW4 dataset to calculate the proportional distribution of livestock across municipalities. These proportions were then applied to each year's national total from FAOSTAT, allowing for the generation of city-level reference data. These refined city-level statistics are then employed to derive pixel-level statistics, which are subsequently overlaid with theoretical suitability masks. Utilizing these masked pixel-level statistics, a stratified sampling approach is implemented, wherein mapping features are incorporated into the random forest regression for training purposes. This yields preliminary mapping results, which are further refined using the corrected country-level statistics.

### 125 3.1 Extracting theoretical suitable masks

As for the multiple spatial resolutions of datasets shown in Table 2 and 3, all the datasets were reprojected and resized to 5 km. Specifically, the resolution of land cover maps were reduced by calculating the percentage of grassland and impervious within the 5-km grids for grazing and captive livestock, respectively. To generate theoretical suitable masks, the land cover percentage maps were further masked with population density less than 250,000 people per km<sup>2</sup> and elevation less than 5600 m. Since the statistics cover up to 1961, for population dataset before 2000, we used the map of 2000 instead. Meanwhile, the land cover masks before 1982 were estimated from 1982. Afterwards, the pixel-level livestock intensities were obtained based on the annual population and land cover products, as well as country-level livestock numbers provided by FAOSTAT.

### 3.2 Feature construction and sampling

Datasets listed in Table 3 were used for the mapping feature construction in this study. Specifically, the vegetation variables were not used for captive livestock mapping. Datasets with multiple imagery during the year, i.e., Climate & Soil and vegetation, were mosaicked using the average strategy. It should be noted that, due to constraints imposed by varying feature

Figure 1. The annual livestock mapping workflow.

availability, the mapping features for different years may vary. In instances where relevant feature data sources are lacking for specific years, corresponding features will be omitted. For example, in the case of sheep mapping for the year 1961, only variables of terrain (elevation and slope) and climate & soil (precipitation, minimum temperature, maximum temperature, wind-speed at 10m, and soil moisture) were utilized as mapping features.

Due to the utilization of the Random Forest model in this study for estimating livestock spatial distribution, the initial step involves the collection of training and validation samples. Given the relatively coarse resolution of the country-level FAOSTAT dataset, the GLW4 datasets were employed to refine the city-level livestock distribution basis. Specifically, the total livestock counts within each city were recalibrated at the city level based on the proportional representation of livestock quantities across cities in the GLW4 datasets. Leveraging these recalibrated statistics, a stratified sampling approach was then implemented. Drawing upon the results of the theoretical suitability masks, this research employed a stratified sampling approach based on the pixel-level livestock density derived from the recalibrated city-level statistics. Given the differences in population size and distribution range among livestock species, we adopted species-specific stratification intervals. For example, for ducks, whose densities tend to be high and spatially heterogeneous, we used a stratification interval of 500 heads per hectare grid cell; for horses, a finer interval of 1 head was applied. Each stratum was randomly sampled, and approximately 20,000 training samples per year were selected for each livestock category.

### 3.3 Livestock mapping with Random Forest regression

For each category of livestock, we embarked on training Random Forest regression models for annual distribution mapping. In this process, 70% of the sample set was allocated for training, while the remaining 30% was reserved for validation. The parameter settings for the Random Forest model were as follows: 1) the number of variables per split was set as 5; 2) the number of decision trees was set as 200; 3) other parameters were set as the default value of the GEE. Based on the outcomes of this model training, we were able to derive initial livestock density distribution estimates for different livestock categories across various years using the feature construction results.

Furthermore, we conducted optimization of the preliminary estimates using city-level statistical data. For each city, the procedure involved computing the livestock quantity (denoted as  $Num_p$ ) based on the model's density estimate output and comparing it to the reassigned city-level livestock count (denoted as  $Num_s$ ). The resulting ratio  $scale = Num_s/Num_p$ , was used to rescale the livestock density estimate within the city, yielding the final optimized livestock density spatial distribution map. Finally, relying on the density spatial distribution results, the quantity distribution maps for different livestock in various years were developed.

### 165 3.4 Evaluation

The evaluation of the AGLW dataset was conducted from three perspectives: model level, finer-scale statistic level, and pixel level. Firstly, the Random Forest regression model, trained with 70% of the sample dataset collected in Section 3.2, was evaluated using the remaining 30% of the sample dataset. The validation samples were used to calculate correlation coefficients (r) to assess the model's accuracy and reliability. Secondly, to verify the reliability of the AGLW dataset at a finer administrative scale, we used annual statistics from seven typical provinces/states and China county-level statistics. These statistics were collected from different national statistical departments for various states/provinces and county-level livestock numbers (Table 4). Meanwhile, since the statistics are annually collected, the finer-scale statistic level evaluation allowed us to assess whether the AGLW dataset accurately captured annual changes in livestock numbers. Thirdly, for pixel-level evaluation, the GLW2, GLW3, and GLW4 datasets were introduced. We focused on the overlapping regions of GLW and AGLW datasets for the years 2006, 2010, and 2015. For each year and each type of livestock, 50,000 random sample points were generated. By conducting linear regression on the livestock density values from GLW and AGLW at these sample points, we calculated the value of r to quantify the consistency between the AGLW and GLW2/3/4.

## 4 Results

## 4.1 Spatial and temporal characteristics of global livestock dynamics

Figure 2 shows the global spatio-temporal distribution dynamics of livestock measured in this study, including cattle, buffaloes, sheep, goats, horses, pigs, chicken, and ducks. Utilizing 1961-2021 FAOSTAT and suitability masks, these annual global maps were developed with mapping features and Random Forest regression, and corrected to match refined city-level statistics.

According to the predicted spatial distribution presented in Figure 2, the highest densities of cattle are located in northwest and northeast of India, northern Europe, middle and east of Africa, and South America. Some scattered regions of high density are distributed in Middle Asia, north of Italy, the central United States, and north of New Zealand. The global distributions of goats and sheep are similar, with dense concentrations found in northwest China, southern India, the Middle East, central and northern Africa. However, in the United Kingdom and southern Oceania, sheep are more densely distributed compared to goats. Horses also have a widespread global distribution, although their density is lower compared to cattle and sheep. Areas with higher densities are primarily located in Mongolia, Ethiopia, Mexico, and the state of Kentucky in the United States.

Figure 2. The gridded livestock of the world (AGLW) dataset (take the density maps of 1961/1981/2001/2021 as examples).

The aggregations of pigs are primarily observed in China, with additional concentrations noted in the middle of the United States. Europe countries (e.g. Denmark, Germany, the Netherlands, France, Spain, and Italy), Africa countries (e.g. Malawi,

Nigeria, and Burundi), and regions in South America (e.g. the middle of Chile, the southern part of Brazil, and the northwest of Colombia), also demonstrate notable concentrations of pigs. Globally, chickens exhibit dense distributions across various regions and are intricately linked to population density. Notable aggregations of chickens are particularly evident in eastern and central Asia, such as China, India, Pakistan, and Iran, and across Europe as a whole. Along the coasts of Africa, chicken densities are also notably high. Similar distributions can also be observed in Central and South America. In the United States, chickens are predominantly distributed in the southeastern region, while the western region is primarily dominated by California. Buffaloes and ducks, on the other hand, are relatively less common compared to the aforementioned types of livestock. Buffaloes exhibit high densities across regions spanning India, Egypt, and southwest China. Ducks are predominantly distributed across the middle and eastern regions of China, Southeast Asia (e.g., Vietnam, Bangladesh, Malaysia, Indonesia), and the western part of France, a distribution closely associated with rice cultivation and the fishing industry.

Unlike the existing global livestock mapping dataset, the dataset constructed in this study can reflect the annual distribution changes of global livestock. We can observe the continuous expansion of almost all livestock distributions over the past 60 years. While this is partly due to the increasing number of countries covered by FAOSTAT, the expansion pattern can also be clearly observed within the same country. For instance, rapid expansions in the distribution of chicken, ducks, and pigs can be observed in China since the 1980s. This expansion is not only related to the continuous population growth in China but also closely linked to the rapid economic development in the region. In contrast, we can observe that some traditional horse-rearing countries, such as Poland, have experienced a decline in production since 1960. This change is associated with the decreasing importance of horses in transportation, agriculture, and other industries as they are gradually replaced by motor vehicles (Goran and Jelisavka, 2014). Meanwhile, for traditional cattle and sheep rearing countries, such as India for cattle and New Zealand for sheep, the fluctuations in their numbers are not significant.

To provide a detailed comparison of livestock density captured by the AGLW dataset with existing GLW dataset, seven typical provinces/states known for their livestock raising were selected. These include cattle in Texas, United States; chickens in California, United States; horses in Kentucky, United States; pigs in Henan, China; buffaloes in Guangxi, China; goats in Inner Mongolia, China; and sheep in New South Wales, Australia. The zoomed mapping details of the AGLW dataset for the year 2015 were compared with the GLW4 dataset, as illustrated in Figure 3. The comparison demonstrates that the livestock density distribution in the AGLW dataset is similar to that in the GLW4 dataset, with notable improvements in thematic details. However, in areas with low livestock density, the differences between the two datasets are minimal. Specifically, Figure 3 showcases the overall similarity and the detailed differences in spatial distribution between the AGLW and GLW4 datasets. This can be attributed to the use of impervious layer construction for creating suitable masks for chickens and pigs in the AGLW dataset, as seen in Figures 3d and 3e. Additionally, Figure 3f highlights a similar issue, which may arise from the use of different land cover products in AGLW and GLW4. The differences in grassland distribution result in varying suitable masks between the two datasets, thereby leading to differences in the final spatial distribution of livestock density. Overall, the comparison results shown in Figure 3 underscore the effectiveness of the AGLW dataset in capturing finer-scale and longer-term livestock spatial distribution globally, while maintaining a high degree of consistency with established datasets like GLW4.

**Figure 3.** Spatial distribution comparison with GLW4 of livestock density in typical regions. a) Cattle in Texas, United States; b) buffaloes in Guangxi, China; c) sheep in New South Wales, Australia; d) chickens in California, United States; e) pigs in Henan, China; f) goats in Inner Mongolia, China; g) horses in Kentucky, United States.

## 4.2 Accuracy assessment

# 4.2.1 Model level evaluation with internal cross

The Random Forest regression mapping model was trained using 70% of the annually generated samples for each category of livestock. The remaining 30% of the samples were reserved for model-level evaluation. For each species of livestock per year, the correlation coefficient (r) was calculated based on the mapping results and sample values. The results indicate a moderate to high correlation across all livestock species, with average r values ranging from 0.54 to 0.73. According to the evaluation results shown in Figure 4, cattle exhibit a slightly lower correlation, suggesting a moderate agreement between the mapping results

and the validation samples. In contrast, goats and horses show higher correlations, reflecting a stronger consistency in density distribution. This variation in model performance across livestock species can be partly attributed to differences in their spatial distribution patterns and management systems. Cattle, as a grazing species, are often raised in extensive and environmentally heterogeneous systems, making their spatial patterns more diffuse and harder to predict accurately. In contrast, horses are typically managed in more spatially concentrated settings, leading to more spatially clustered distributions and better model fit. Moreover, Figure 4 illustrates that sheep and pigs have relatively higher interannual uncertainty. However, it is important to note that the validation samples used in the model evaluation were estimated using the sampling method introduced in Section 3.2. Therefore, they do not measure the correspondence between the predicted animal densities and the actual ground truth values. To achieve more reliable validation results, this study further utilized finer-scale statistics and existing global datasets for statistic and pixel level validations.

Figure 4. Internal cross model evaluation results for each specie of livestock. Dots represent correlation coefficients (r) between mapping results and validation samples for each year.

## 4.2.2 Finer-scale statistic level evaluation

The AGLW dataset was developed using annual country-level FAOSTAT statistics. To evaluate the effectiveness of the dynamics presented by AGLW at a finer scale, the annual livestock numbers of typical states/provinces were validated with annual statistics shown in Table 4 (Figure 5a). According to the comparison results, the annual numbers reflected in this study closely match the province/state level statistics collected by statistical offices in different countries (r = 0.98). This high level of concordance underscores the reliability of the AGLW dataset in reflecting annual livestock number fluctuations. In addition to province/state level validation, we further examined the dataset's accuracy at the county level within China. For this purpose, we obtained county-level statistics for the years 1990, 2002, 2007, 2012, and 2017 and compared these statistics to those derived from the AGLW dataset (Figure 5b). Similarly, the data from this study and the China Statistical Yearbook show good consistency at the county level (r = 0.79). It suggests that the AGLW dataset, despite being constructed from national-scale statistics, is capable of accurately depicting livestock dynamics at much finer scales. These validation results illustrate

the robustness of our mapping method across years and its applicability in generating detailed spatiotemporal distributions of various livestock species at city and county levels. The ability to achieve finer-scale consistency using national-level input data highlights the efficacy of our approach and its potential utility in regional and local livestock management and planning.

**Figure 5.** Livestock number comparisons with a) state/province level and b) county level statistics. Dots represent the number of livestock at administrative levels based on AGLW and statistical records.

### 4.2.3 Pixel level evaluation with GLW

For the pixel-level evaluation of the AGLW dataset, 50,000 points were randomly generated for GLW2, GLW3, and GLW4, respectively. The annual maps of AGLW for the years 2006, 2010, and 2015 were then used to complete this pixel-level assessment. According to the results shown in Figure 6, the consistency between the AGLW and GLW datasets is relatively high, with correlation coefficients (r) of 0.73, 0.78, and 0.83, respectively. Overall, when compared to the GLW datasets, the AGLW data tend to show an overestimation in low-density areas and an underestimation in high-density areas. Specifically, the highest consistency is observed between AGLW and GLW4, followed by GLW3, and then GLW2. This trend can be attributed to the use of GLW4 data in constructing the refined city-level statistics within this study, as well as the superior spatial details present in the GLW4 dataset.

The high correlation coefficients indicate a strong agreement between the AGLW dataset and the GLW datasets, validating the accuracy and reliability of the AGLW maps at a pixel level. This demonstrates the effectiveness of our methodology in capturing livestock distribution patterns with considerable detail and precision. By processing both model level, statistic level, and pixel level evaluation, our approach ensures that the AGLW dataset remains robust across different spatial scales and densities. This makes it a valuable tool for both global and localized analyses, offering insights that are critical for sustainable livestock management and planning.

**Figure 6.** Livestock density comparison with GLW datasets. Dots represent the pixel-level livestock density at sampled locations from both AGLW and GLW datasets.

#### 5 Discussion

Based on the evaluation of product intercomparison, accuracy, and consistency, it is evident that the annual global livestock mapping products spanning from 1961 to 2021 exhibit reliability, underscoring the effectiveness of the mapping framework developed in this study. Utilizing the 61-year livestock mapping results, this dataset holds considerable potential for socioeconomic, environmental, and health studies, particularly for long-term applications on a global and continental scale.

However, despite the strengths of our mapping framework and dataset, certain deficiencies persist. First, due to the challenges associated with accessing mapping features for historical years, some mapping processes only utilized a subset of training features in certain years. For instance, since the difficulty of obtaining vegetational features prior to 1980, livestock mapping of this period relied on terrain and climate & soil variables. Consequently, this limitation may introduce uncertainty of the results for these years. In order to quantify the uncertainty introduced by these data acquisition limitations, we conducted sensitivity analyses for the 2015 livestock mapping, exploring the impact of different feature inputs on mapping outcomes. The values of r for different livestock species were calculated using various feature groups (Figure 7), which include all features listed in Table 3 and subsets excluding specific categories: anthropogenic features (Group 1), topography features (Group 2), climate and soil features (Group 3), and vegetation features (Group 4). According to the correlation analysis results using validation samples introduced in Section 3.2, the mapping results that utilized all features achieved the highest r values, while Group 3 (excluding climate and soil features) had the lowest r values. This indicates that, when all features are available, using the complete set for livestock mapping is the optimal approach. Although vegetation features were not used in the mapping before 1980 due to data availability constraints in this study, Figure 7 shows that this omission had a minimal impact on the final mapping results. In contrast, the exclusion of climate and soil features has a more pronounced effect on final mapping results. The relatively minor influence of anthropogenic and vegetation features may be attributed to spatial correlations between human activity indicators and the suitable mask (e.g., impervious surface layers), and the use of temporally static historical data before the years of 2000 and 1980.

**Figure 7.** The correlation coefficient comparison for Random Forest regression models with different input features. Dots represent correlation coefficients (r) obtained from models trained with different feature combinations.

(All: all features; Group1: except anthropogenic features; Group2: except topography features; Group3: except climate & soil features; Group4: except vegetation features)

To further investigate the role of different input features and their influence on mapping outcomes, we performed Partial Dependence Plot (PDP) analyses using two representative livestock species: cattle and ducks. These species were selected due to their differing habitat preferences and spatial distributions, providing complementary perspectives on feature importance. The PDP results (Figures S1 and S2) reveal several consistent patterns, suggesting common influential factors of livestock distribution. Notably, population density, precipitation, and soil moisture show positive associations with predicted livestock density for both cattle and ducks. This highlights the importance of human activity and water availability in shaping livestock distributions. For instance, cattle and ducks both exhibit higher predicted densities in regions with greater population, suggesting the influence of demand-side factors such as local consumption and infrastructure accessibility. Additionally, elevation and wind speed at 10 m consistently show negative contributions across both PDPs, indicating a general preference for lower-elevation and less windy environments, which are typically more suitable for animal husbandry. Vegetation features (e.g. total number of valid vegetation cycles with peak) also display positive relationships with livestock density (Parente et al., 2025). These PDP results reinforce the rationale for selecting a comprehensive set of input features wherever data availability permits.

Second, constrained by the global scale of livestock statistics, this study currently utilizes country-level statistics for mapping and refines them to the city level using the GLW4 dataset. Accessing statistical data directly at finer levels would lead to more refined and accurate mapping outcomes for livestock dynamics. Here, leveraging county-level livestock statistics from the validation data introduced in Section 2.2.1, we implemented pig mapping in China for the years of 1990 and 2017 based on the mapping method described in this paper. By utilizing annual county-level statistics, the distribution of statistical data dynamics was allocated at a finer scale. This approach, compared to using country-level statistical data, mitigates the uncertainty introduced by the Random Forest regression and allows for a more accurate depiction of livestock dynamics (Figure 8). This

finer scale mapping provides a clearer understanding of the spatial and temporal changes in livestock populations, leading to better-informed decision-making in livestock management and policy formulation.

Figure 8. Pig mapping in China with county-level statistics for the years of a) 1990 and b) 2017.

To enable global-scale and long-term consistency, our study adopted a proportion-based downscaling approach using the GLW4 dataset to redistribute FAOSTAT national totals at city level. While this method assumes relative stability in subnational livestock distributions across time, which may introduce uncertainty in dynamic regions, it is supported by previous large-scale studies (Theobald et al., 2020; Van Boeckel et al., 2019). Nevertheless, we recommend that users exercise caution when applying these data in regions with known subnational shifts in production systems. In the future, with the continued refinement of livestock statistics, we will be able to enhance the precision and utility of livestock maps within this framework. Additional province, city, or county-level statistical data available globally would be collected to create regionally refined and multi-scale mapping products. Concurrently, building upon this foundation, the potential of land system modeling algorithms would be explored to create a more extensive livestock mapping dataset with finer temporal and spatial resolutions. Such efforts will bolster domains linked to this dataset, such as socio-economic, environmental, and health research.

### 325 6 Data availability

The AGLW dataset generated in this study can be publicly accessed at https://doi.org/10.5281/zenodo.11545701 (Du et al., 2025). The maps are grouped by species, including cattle, buffaloes, horses, sheep, goats, pigs, chickens, and ducks. For each species, annual maps during 1961-2021 were provided at the 5 km spatial resolution.

### 7 Conclusions

Global livestock dynamics provide key variables for a wide range of Earth system science studies, including land cover and use change analysis, public health, ecosystem monitoring, and sustainable development. Leveraging the strengths of FAOSTAT and the Random Forest regression model, this research developed the first annual long-term global livestock maps of AGLW from 1961 to 2021, and addressed the previous limitations in spatial and temporal continuity and resolution found in existing products. The resulting annual maps of AGLW exhibit a high degree of consistency with validation samples, province/state statistics, county statistics, and existing global data products, with correlation values of 0.54-0.73, 0.98, 0.79, and 0.73-0.83. The spatio-temporal dynamics presented by AGLW reveal the overall expansion of livestock globally over the past six decades, alongside localized fluctuations in specific species and regions, such as the significant increase in pig stock in China and the decline in horse stock in Poland. Therefore, the AGLW offer a vital resource that enhances our understanding of global livestock dynamics, thereby informing policy decisions, guiding sustainable agricultural practices, and fostering resilience in ecological and human systems.

*Author contributions.* L.Y. conceived the experiment, Z.D. conducted the experiment, all authors analyzed the results, all authors reviewed the manuscript.

Competing interests. The authors declare that they have no conflict of interest.

Acknowledgements. This work was supported by the National Key R&D Program of China (2024YFF1307600), National Natural Science 345 Foundation of China (42201367), Liaoning Province Natural Science Foundation under Grant (2024-BSBA-04), and Fundamental Research Funds for the Central Universities (DUT23RC(3)064).

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
