# Peer review of "Annual global grided livestock mapping from 1961 to 2021"

_Earth System Science Data, 2025_

## Author Comment (AC1)

We would like to thank the editor and reviewers very much for the valuable comments and suggestions that greatly helped us to improve the manuscript. Thank you very much for your time and efforts. In this revised version, we have addressed all reviewer comments in detail. Major revisions include: (1) refining the discussion on data uncertainty and variable importance, especially regarding vegetation features and anthropogenic drivers; (2) incorporating Partial Dependence Plot (PDP) analyses to improve the interpretability of the models; (3) clarifying the rationale behind the use of suitability masks and addressing concerns about land cover assumptions; (4) explaining the variation in prediction accuracy across species. We believe these revisions have substantially strengthened the scientific rigor, clarity, and transparency of the manuscript.

**Reviewer #1:**
**_Comment 1._** _The authors rely on FAOSTAT's national-level livestock statistics as the primary data source for mapping. While these data span a long temporal range (1961–2021), their spatial resolution is generally coarse. Deriving gridded datasets primarily based on these national statistics may introduce substantial spatial uncertainty, as livestock distributions exhibit strong intra-national heterogeneity (https://doi.org/10.1016/j.oneear.2023.08.012; https://doi.org/10.1016/j.rse.2019.111301). And this issue could be particularly pronounced in large, transhumant livestock nations such as the United States, China, Brazil, and India._

**Response:** Thanks for the comment. We agree that national-level statistics, such as those from FAOSTAT, are spatially coarse and do not capture intra-national heterogeneity. To mitigate this limitation, we implemented a spatial downscaling approach that uses the GLW4 dataset as a baseline to proportionally allocate FAOSTAT's national totals to the city level. Specifically, we calculated city-level livestock proportions from the GLW4 density maps and used these proportions to rescale the annual national totals from FAOSTAT. This method enables us to generate subnational reference distributions for both sample generation and model correction (as the reviewer kindly pointed out in Comment 3).

We acknowledge that this method inherits the assumption of spatial stability in livestock distributions over time. To evaluate this, we conducted validation using multi-year subnational statistics (Figures 5 and 6), which showed that the resulting spatial-temporal patterns aligned well with observed dynamics at both provincial and county levels. For further details and supporting literature, we kindly refer the reviewer to our response to Comment 3.

**_Comment 2._** _As noted in the discussion (Lines 281–290, Figure 8), the authors indicate that adopting finer-scale livestock statistics (e.g., municipal or county-level) is one of the most effective methods to reduce uncertainties. In fact, numerous studies have already leveraged such high-resolution data to develop regional spatial datasets, such as https://www.nature.com/articles/s41597-024-03072-y; https://doi.org/10.5194/essd-13-515-2021. A recent study even compiled over 50,000 fine-scale records for global_

_livestock mapping (https://doi.org/10.21203/rs.3.rs-6201916/v1). Compared to these efforts, what advantages does this study offer in uncertainty control?_

**Response:** We thank the reviewer for pointing us to the valuable references, which represent significant progress in high-resolution livestock mapping. We fully agree that finer-scale statistical records (e.g., municipal or county-level) are essential for reducing uncertainty in livestock distribution estimates. However, while such fine-resolution datasets are increasingly available for selected regions and recent years, it remains very difficult to obtain globally consistent, spatially harmonized, and temporally comparable high-resolution statistics across multiple decades. Most available fine-scale data, even within a single country, are limited in temporal coverage, vary in format and definitions, and are difficult to standardize for use in long-term global mapping. In contrast, FAOSTAT provides the globally consistent livestock statistics from 1961 onward, albeit at the national level.

In this context, the main advantages of our study in terms of uncertainty control are as follows: (1) By using a uniform, global input source (FAOSTAT) and consistent modeling procedures, we ensure that interannual changes in livestock distribution are comparable and not confounded by inconsistent data sources. (2) Our study not only acknowledges uncertainty but also evaluates it at multiple levels—model performance (Fig. 4), spatial consistency with fine-scale statistics (Fig. 5, Fig. 8), and feature sensitivity (Fig. 7). This multi-level assessment helps identify which inputs and assumptions are driving spatial or temporal uncertainty. (3) While our current product is based on national statistics, we demonstrated in Figure 8 that integrating finer-scale statistics (e.g., Chinese county-level pig data) improves spatial allocation accuracy. This shows that our framework can flexibly incorporate finer data wherever available, without losing global scalability.

In summary, while regionally detailed maps are valuable, our contribution lies in producing the first globally consistent, annually gridded livestock dataset covering 61 years, with built-in mechanisms for uncertainty evaluation and integration with finer-scale data in future updates. We believe this temporal and methodological robustness offers a distinct and complementary advantage to existing efforts.

**_Comment 3._** _The authors mention using GLW4 to downscale FAOSTAT's national statistics to municipal (city) scales (Lines 117–119), yet the specific methodology remains unclear. Is the process based on calculating municipal proportions from GLW4 data and then scaling national totals by these proportions? If so, this approach may inherit significant uncertainties, as municipal proportions can vary substantially over time._

**Response:** Thanks for the comment. Yes, the reviewer is correct in interpreting our approach: we used the livestock density distributions from the GLW4 dataset to calculate municipal-level proportions, and then applied these proportions to redistribute FAOSTAT national totals to a finer administrative scale for sample generation and model correction. We have clarified the specific processing steps in the revised Methods section as below (Lines 117-121):

"…FAOSTAT serves as the primary input for country-level statistics and acts as

the basis for corrections. To refine these statistics to the city level, we used the GLW4 dataset to calculate the proportional distribution of livestock across municipalities. These proportions were then applied to each year's national total from FAOSTAT, allowing for the generation of city-level reference data to guide stratified sampling and to rescale model outputs…"

We acknowledge the uncertainty that this method may introduce, particularly due to the assumption that subnational distributions remain temporally stable. To assess its validity, we conducted further validation using multi-year subnational statistics (Figures 5 and 6). These comparisons demonstrate that the resulting time series preserve reasonable temporal dynamics, despite the use of a single-year GLW4 dataset as a reference.

Moreover, this assumption of spatial stability is consistent with many previous studies. For example, Van Boeckel et al. (2019) [https://doi.org/10.1126/science.aaw1944] used 2010 GLW data to assess global antimicrobial resistance in livestock without altering the spatial distribution across years, implying that the baseline livestock distribution was relatively stable for cross-year comparisons. Similarly, Theobald et al. (2020) [https://doi.org/10.5194/essd-12-1953-2020], in their study on global human modification from 1990 to 2017, employed a static livestock layer over multiple years, treating livestock pressure zones as relatively stable spatial variables. These examples indicate that the spatial distribution of livestock, tends to remain coherent over time, especially when constrained by agroecological, infrastructural, and cultural factors.

To ensure transparency, we have further discussed this potential source of uncertainty in the revised Discussion section and explicitly cautioned users regarding the assumptions made in the temporal allocation of livestock distributions (Lines 315-319):

"To enable global-scale and long-term consistency, our study adopted a proportion-based downscaling approach using the GLW4 dataset to redistribute FAOSTAT national totals at city level. While this method assumes relative stability in subnational livestock distributions across time, which may introduce uncertainty in dynamic regions, it is supported by previous large-scale studies (Theobald et al., 2020; Van Boeckel et al., 2019). Nevertheless, we recommend that users exercise caution when applying these data in regions with known subnational shifts in production systems."

***Comment 4.*** *Based on the difference in feeding systems, authors categorize animals into "grazing livestock" (e.g., buffalo, cattle, goats, horses, sheep) and "captive livestock" (e.g., chickens, ducks, pigs), and assume grazing species inhabit grasslands while captive species are confined to impervious surfaces (Lines 83–85). This assertion appears questionable, as intensively raised animals often occupy peri-urban or rural agricultural lands (https://doi.org/10.1016/j.oneear.2023.08.012).*
**Response:** Thanks for this important observation and for recommending reference. In direct response to this reference—and since the study focuses on pigs—we have chosen to illustrate our clarification using pig mapping as an example. Although we used

impervious surfaces as a component of the suitability mask for livestock, we concurrently applied a population density to remove dense urban centers from the suitable zones (Please kindly check Figure 1). This approach was designed to retain peri-urban and rural impervious areas, which are typically associated with livestock farming operations.

To demonstrate this approach, we added a representative figure showing pig density mapping in Jiangsu Province, China (see figure below). In the left panel, areas with high pig density in 2021 are shown, and it is clear that densely populated coastal cities (e.g., cities on the east coast) were excluded from the suitability area. In the right panels, a representative pig farm is outlined in a white circle. This farm is located in a peri-urban area and is mapped as impervious surface (red pixels) in the land cover product. Notably, it is surrounded by rural agricultural lands (orange pixels), and its location aligns with a high-density zone in the pig distribution map. This example proves that our strategy is able to capture livestock production zones located in peri-urban agricultural landscapes.

[Figure]

***Comment 5.*** *The discussion is not very adequate. (1) For instance, the claim that vegetation omission minimally impacts predictions (Lines 280–281) is counterintuitive. What underlying reasons justify this assertion? Have other studies observed similar patterns? Is it premised on the assumption that grasslands or impervious surfaces serve as "theoretical suitable masks" for livestock distribution (Lines 83–85)? (2) Additionally, Figure 4 shows marked disparities in prediction accuracy across species (notably lower for cattle and higher for horses). What factors explain these variations?*

**Response:** Thanks for the comment. Following the suggestions of both reviewers, we have expanded the discussion to more fully address the uncertainty of our mapping product and the contributions of different predictor groups. Regarding the two specific questions raised in this comment, we respond as follows.

(1) Based on our sensitivity analysis (Figure 7), omitting vegetation-related features led to only a marginal decrease in prediction accuracy, especially when compared to the exclusion of climate and soil variables. Several factors may explain

this result. First, in global-scale models where spatial resolution is relatively coarse, vegetation indices—especially those derived from satellite imagery—tend to be temporally noisy and strongly affected by seasonality and land use dynamics, reducing their predictive reliability compared to more stable features such as climate and terrain.

Second, as the reviewer kindly pointed out, our use of land-cover-based "suitability masks" (e.g., impervious surfaces and grasslands) already filters the spatial domain in a way that may absorb some vegetation-related variation, thereby reducing the marginal effect of vegetation features. A similar pattern was observed in a recent study. Parente et al. (2025) noted: "This may be explained by the fact that we only consider the fraction of forested land within areas suitable for livestock." (https://doi.org/10.21203/rs.3.rs-6201916/v1).

To further analyze the contribution of individual features to prediction outcomes, we followed the other reviewer's recommendation and performed a Partial Dependence Plot (PDP) analysis. The results, presented in Supplementary Figures S1 and S2, show a positive association between vegetation features and livestock density. This conclusion has also been proved in the Figure 8 of Parente et al. (2025). The clarification has been added in the section of Discussion (Lines 290-305):

"…The relatively minor influence of anthropogenic and vegetation features may be attributed to spatial correlations between human activity indicators and the suitable mask (e.g., impervious surface layers), and the use of temporally static historical data before the years of 2000 and 1980."

"To further investigate the role of different input features and their influence on mapping outcomes, we performed Partial Dependence Plot (PDP) analyses using two representative livestock species: cattle and ducks. These species were selected due to their differing habitat preferences and spatial distributions, providing complementary perspectives on feature importance. The PDP results (Figures S1 and S2) reveal several consistent patterns, suggesting common influential factors of livestock distribution…Vegetation features (e.g. total number of valid vegetation cycles with peak) also display positive relationships with livestock density (Parente et al., 2025)."

(2) The observed variation in accuracy (e.g., lower for cattle, higher for horses) likely reflects differences in the ecological characteristics and management systems of each species. Grazing animals like cattle are typically associated with extensive pastoral systems that are more spatially diffuse and environmentally constrained, making them more difficult to model accurately. In contrast, horses tend to be concentrated around built environments (e.g., stables, equestrian facilities) and are often managed in more predictable locations, leading to higher model performance. Similar patterns were also found in Ehrmann et al. (2025), where prediction accuracy (measured by $R^2$) was significantly higher for horses ($R^2 = 0.530$) than for cattle ($R^2 = 0.437$). We have included this explanation in the updated model accuracy assessment section of the manuscript (Lines 234-238):

"This variation in model performance across livestock species can be partly attributed to differences in their spatial distribution patterns and management systems. Cattle, as a grazing species, are often raised in extensive and environmentally heterogeneous systems, making their spatial patterns more diffuse and harder to predict

accurately. In contrast, horses are typically managed in more spatially concentrated settings, leading to more spatially clustered distributions and better model fit."

---

## Author Comment (AC2)

We would like to thank the editor and reviewers very much for the valuable comments and suggestions that greatly helped us to improve the manuscript. Thank you very much for your time and efforts. In this revised version, we have addressed all reviewer comments in detail. Major revisions include: (1) refining the discussion on data uncertainty and variable importance, especially regarding vegetation features and anthropogenic drivers; (2) incorporating Partial Dependence Plot (PDP) analyses to improve the interpretability of the models; (3) clarifying the rationale behind the use of suitability masks and addressing concerns about land cover assumptions; (4) explaining the variation in prediction accuracy across species. We believe these revisions have substantially strengthened the scientific rigor, clarity, and transparency of the manuscript.

**Reviewer #2:**
*Comment 1. The methods section lacks clarity in certain areas, particularly regarding the stratified sampling approach. The manuscript does not clearly describe how stratified sampling was implemented (L140-L146). This information is critical, as it directly influences the composition of the training dataset and consequently affects the accuracy and reliability of the global predictions. I recommend that the authors provide a more detailed explanation of the sampling procedure, including the criteria for stratification and how the strata were defined and selected.*
**Response:** Thanks for the comment. We agree that the stratified sampling strategy plays a crucial role in ensuring representative training data and improving model accuracy. We have revised the manuscript to clarify the stratification criteria and sampling intervals. Specifically, the stratification was based on pixel-level livestock density values derived from the recalibrated city-level statistics. Given the wide variation in livestock abundance across different species, we adopted species-specific stratification intervals. For instance, for ducks, which have high population densities and wide spatial variability, we used a stratification interval of 500 heads per grid cell. In contrast, for horses, a smaller interval of 1 head was used. Within each stratum, samples were randomly selected to ensure sufficient representation across density gradients. We have included this information in the revised Methods section (Lines 147-151) accordingly:

"Given the differences in population size and distribution range among livestock species, we adopted species-specific stratification intervals. For example, for ducks, whose densities tend to be high and spatially heterogeneous, we used a stratification interval of 500 heads per hectare grid cell; for horses, a finer interval of 1 head was applied. Each stratum was randomly sampled, and approximately 20,000 training samples per year were selected for each livestock category."

*Comment 2. The causal relationships between the predictors and the response variable warrant further clarification. In this study, the authors used a range of environmental and anthropogenic factors to predict livestock density (Fig 1). For predictors with limited historical data, such as population, the authors applied year-2000 values to years before 2000 and found that population had little influence. This conclusion seems counterintuitive. Unlike wildlife, livestock is more likely to be influenced by human*

*management. Therefore, one would expect population density to be an important predictor. However, in this study, soil and climate variables were found to be more influential (fig 7). This may reflect correlations rather than causal mechanisms. A comparison between the spatial patterns of cattle or sheep and population density (https://hub-worldpop.opendata.arcgis.com/content/WorldPop::global-1km-population-total-grid-2000-2020/about) suggests that a strong spatial association likely exists. I think that the lack of observed influence in the model may be due to two reasons: (1) errors or bias introduced during stratified sampling (as noted in comment 1); and (2) potential multicollinearity among predictors. If population is indeed an important factor, I think the authors to revisit its treatment carefully. In addition, I strongly recommend including partial dependence plots or similar visualizations to show how each predictor relates to the response variable.*

**Response:** We appreciate the reviewer's comment regarding the interpretation of predictor influence, particularly the role of population density in livestock distribution modeling. We acknowledge that livestock is highly influenced by human activities, including population distribution, market access, and infrastructure. However, due to the lack of globally available historical population data prior to 2000 at consistent resolution, we used the year-2000 WorldPop layer as a proxy for years before 2000. We agree that this temporal mismatch could introduce uncertainty, especially in regions where population patterns have changed significantly. We have revised the Discussion section to clarify this limitation (Lines 290-293):

"The relatively minor influence of anthropogenic and vegetation features may be attributed to spatial correlations between human activity indicators and the suitable mask (e.g., impervious surface layers), and the use of temporally static historical data before the years of 2000 and 1980."

The observed limited contribution of population in our feature importance ranking (Fig. 7) may be attributed to population being partly spatially correlated with our suitability masks (especially impervious surface). To better illustrate the marginal effects of individual predictors and improve interpretability, we have now included partial dependence plots (PDPs) for all mapping features and two representative livestock types (cattle and ducks), as the reviewer kindly suggested. These new plots are added as a supplementary figure (Fig. S1 and Fig. S2), and referenced in the Discussion section (Lines 294-305):

"To further investigate the role of different input features and their influence on mapping outcomes, we performed Partial Dependence Plot (PDP) analyses using two representative livestock species: cattle and ducks. These species were selected due to their differing habitat preferences and spatial distributions, providing complementary perspectives on feature importance. The PDP results (Figures S1 and S2) reveal several consistent patterns, suggesting common influential factors of livestock distribution. Notably, population density, precipitation, and soil moisture show positive associations with predicted livestock density for both cattle and ducks. This highlights the importance of human activity and water availability in shaping livestock distributions. For instance, cattle and ducks both exhibit higher predicted densities in regions with greater population, suggesting the influence of demand-side factors such as local

consumption and infrastructure accessibility. Additionally, elevation and wind speed at 10 m consistently show negative contributions across both PDPs, indicating a general preference for lower-elevation and less windy environments, which are typically more suitable for animal husbandry. Vegetation features (e.g. total number of valid vegetation cycles with peak) also display positive relationships with livestock density (Parente et al., 2025). These PDP results reinforce the rationale for selecting a comprehensive set of input features wherever data availability permits."

[Figure]

Figure S1. Partial dependence plots (PDPs) for cattle mapping in 2015. Features include anthropogenic (e.g., population, distance to cities), topographic (elevation, slope), climatic (precipitation, temperature, wind), soil (soil moisture), and vegetation variables (NDVI, green up, senescence, number of cycles).

[Figure]

Figure S2. Partial dependence plots (PDPs) ducks mapping in 2015. Features include anthropogenic (e.g., population, distance to cities), topographic (elevation, slope), climatic (precipitation, temperature, wind), and soil (soil moisture).

***Comment 3.*** *The meaning of the dots in some figures (e.g., figs 4-7) should be clarified in the figure captions.*

**Response:** We have revised the figure captions for Figures 4–7 to explicitly clarify the meaning of the dots. Specifically, in Figures 4 and 7, the dots represent correlation coefficients (r). In Figure 5, the dots indicate the number of livestock. In Figure 6, the dots indicate the pixel-level livestock density. These clarifications are now included in the updated figure captions.

---

## Referee Report (RR1)

The authors made some additions and explanations to the manuscript, but still failed to effectively address several key issues present in the manuscript and raised some new concerns regarding data quality. Specifically:

- 1. As the authors pointed out, one of the key steps of this work was that they generated municipal-level livestock data based on GLW4's grid data and FAOSTAT's national-scale livestock data, and then used it for modeling (Lines 117-119). This approach introduces significant spatial uncertainty: GLW4 only represents the global livestock distribution pattern in 2015, yet the study spans 1961–2021—a period marked by substantial shifts in livestock geography, such as China's livestock industry migrating notably northward between 1978 2014 (https://doi.org/10.1016/j.agrformet.2019.03.022). Consequently, relying on static 2015 data inherently fails to account for these dynamic spatial variations, conflicting with the study's aim to analyze temporal trends. This limitation is corroborated by Figure 6, where validating GLW4 (2015), GLW3 (2010), and GLW2 (2005) against the results reveals progressively declining correlation coefficients (r =  $0.84 \rightarrow 0.78 \rightarrow 0.73$ ), indicating a ~15% decrease in r-values over a decade—a clear signal of spatial reconfiguration that undermines extrapolating 2015 patterns to earlier decades, especially given the 60-year study span. Additionally, how does the author prove the reliability of its earlier data such as those from the 1960s?
- 2. Another key step of this work is that authors categorize animals into "grazing livestock" (e.g., buffalo, cattle, goats, horses, sheep) and "captive livestock" (e.g., chickens, ducks, pigs), and assume grazing species inhabit grasslands while captive species are confined to impervious surfaces (Lines 83-85). This simplification is problematic, as intensively raised livestock (e.g., pigs) frequently occupy peri-urban or rural agricultural lands rather than impermeable surfaces alone (https://doi.org/10.1016/j.oneear.2023.08.012). In the reply letter, the authors cited Jiangsu pig farms to validate this classification, but subsequent checks revealed the mapped "impermeable surfaces" correspond to industrial calcium production facilities (name: 晶诚钙业 Jingcheng Calcium Industry 晶诚钙业 百度地图) rather than pig farms. Given the global prevalence of industrial sites, such misclassifications risk severely compromising data product accuracy.

 $https://map.baidu.com/search/\%E6\%99\%B6\%E8\%AF\%9A\%E9\%92\%99\%E4\%B8\%9A/@13417\\ 359.099172773,3763315.68835945,16.9z/maptype%3DB_EARTH_MAP?querytype=s&da_src=s\\ hareurl&wd=\%E6\%99\%B6\%E8\%AF\%9A\%E9\%92\%99\%E4\%B8\%9A&c=161&src=0&wd2=\%\\ E5\%8D\%97\%E9\%80\%9A\%E5\%B8\%82\%E5\%A6\%82\%E7\%9A\%8B\%E5\%B8\%82&pn=0&sug$

=1&l=18&b=(13417146.36149944,3762923.365705509;13418388.162467439,3763545.1138780 094)&from=webmap&biz\_forward=%7B%22scaler%22:1,%22styles%22:%22sl%22%7D&sug\_f orward=bc0a6b6001c7677833169be2&device\_ratio=1

---

## Author Response (AR2)

We would like to thank the editor and reviewers again for the valuable comments and suggestions that greatly helped us to improve the manuscript. Thank you very much for your time and efforts. In this major revision, we reworked the entire pipeline end-to-end: rebuilt the suitability masks following the reviewer's suggestion, retrained all species models, remapped the full 1961–2021 time series, and redid all validations and downstream analyses (multi-scale checks at county/city/province/state levels and independent comparisons to GLW products). To make limitations explicit, we also added a per-pixel, per-year uncertainty layer that integrates (i) temporal extrapolation with local sample support, (ii) feature completeness by species/year, and (iii) MESS-like environmental novelty. We encourage users to consult the accompanying uncertainty layers when interpreting historical results.

**Comment 1.** (1) As the authors pointed out, one of the key steps of this work was that they generated municipal-level livestock data based on GLW4's grid data and FAOSTAT's national-scale livestock data, and then used it for modeling (Lines 117-119). This approach introduces significant spatial uncertainty: GLW4 only represents the global livestock distribution pattern in 2015, yet the study spans 1961–2021—a period marked by substantial shifts in livestock geography, such as China's livestock industry migrating notably northward between 1978 (https://doi.org/10.1016/j.agrformet.2019.03.022). Consequently, relying on static 2015 data inherently fails to account for these dynamic spatial variations, conflicting with the study's aim to analyze temporal trends. This limitation is corroborated by Figure 6, where validating GLW4 (2015), GLW3 (2010), and GLW2 (2005) against the results reveals progressively declining correlation coefficients ( $r = 0.84 \rightarrow 0.78 \rightarrow$ 0.73), indicating a  $\sim$ 15% decrease in r-values over a decade—a clear signal of spatial reconfiguration that undermines extrapolating 2015 patterns to earlier decades, especially given the 60-year study span. (2) Additionally, how does the author prove the reliability of its earlier data such as those from the 1960s?

**Response:** (1) We thank the reviewer for prompting this important refinement. We acknowledge the substantial difficulty of assembling globally consistent, fine-scale inputs for the 1990s and earlier decades. Consequently, we explicitly recognize that uncertainty is higher for early-period maps (1960s–1990s). Even so, at the global scale the maps still capture the major distributional patterns of different livestock types. To enable careful use of the dataset, this revision introduces a per-pixel, per-year uncertainty layer with the following method (Lines 180-188):

"For uncertainty quantification, we accompany each 5-km annual map with a perpixel uncertainty index  $U \in [0, 1]$ , computed as the mean of three components: temporal extrapolation and sample support, feature completeness, and model applicability. First, temporal extrapolation and sample support combine the normalized distance from the reference year 2015 with a local sample-sparsity score (training points counted within a  $5 \times 5$ -pixel window); larger values indicate greater extrapolation and weaker local support. Second, feature completeness penalizes years/species with missing inputs—years with more available predictors receive lower uncertainty. Third, model applicability adopts a Multivariate Environmental Similarity Surfaces (MESS)

approach widely used in species distribution mapping: for each predictor we compare pixel values to the training 5th–95th percentile range, take the minimum similarity across predictors, and convert it to a 0–1 penalty. Higher U denotes higher uncertainty arising from larger temporal gaps, incomplete features, or extrapolation beyond the training domain."

For uncertainty evaluation results, please kindly check Figure S1:

Figure S1. Livestock mapping uncertainty of AGLW dataset (take maps of 1961/1981/2001/2021 as examples).

In addition, it is important to note that our mapping framework did include time-varying environmental covariates in the Random Forest model. These covariates change over time and can drive some spatiotemporal shifts in the predicted distribution. This means our method is not completely "frozen" to 2015 patterns – it can adjust density based on suitability changes. However, we acknowledge that these indirect adjustments may not fully capture all historical shifts, especially those driven by management and policy. Therefore, we have further discussed this potential source of uncertainty in our previous response (Lines 326-330):

"To enable global-scale and long-term consistency, our study adopted a

proportion-based downscaling approach using the GLW4 dataset to redistribute FAOSTAT national totals at city level. While this method assumes relative stability in subnational livestock distributions across time, which may introduce uncertainty in dynamic regions, it is supported by previous large-scale studies (Theobald et al., 2020; Van Boeckel et al., 2019; Xu et al., 2019). Nevertheless, we recommend that users exercise caution when applying these data in regions with known subnational shifts in production systems."

(2) Our mapping for the 1960s relies on FAOSTAT national livestock totals, which are the official statistics reported by countries. While these are the best available source and provide continuity back to 1960s, we acknowledge that the farther back in time, the more uncertainty may exist in some countries' reported numbers. We assume FAOSTAT's long-term time series is internally consistent and captures the broad trends, but there is an inherent limitation in verifying those 1960s figures on a fine scale. That said, at the country level, the data are as reliable as the FAO sources, and our maps will always match those national totals by construction. Thus, from a macro perspective, the early-year aggregate livestock counts are reliable in our dataset, it is the sub-national distribution of those animals that is uncertain.

For model performance and validation for early years, directly validating the spatial accuracy in the 1960s is extremely difficult, as detailed subnational livestock surveys or maps from that era are generally unavailable globally. However, we have performed several validations that give us confidence in our early-year results' plausibility. Specifically, we chose 7 typical regions for each livestock, and compared our outputs against province- and state-level historical data for a few representative regions (as listed in Table 4 of the paper). Notably, this included cattle in Texas (United States) with data starting from 1969, pigs in Henan (China) from 1978, and buffaloes in Guangxi (China) from 1978. Our results showed a very high correlation (r = 0.97)with these province/state time series, indicating that the temporal fluctuations and general magnitude in those regions are well-captured even in the earlier decades. While this doesn't guarantee pixel-level accuracy, it demonstrates that the overall trends from the 1960s-1970s in those areas are correctly represented by our model. In addition, we performed county-level validation in China from 1990 (the earliest available county data), finding good agreement (r = 0.78 at county scale). By extension, we expect maps of 1960s are not be wildly off in major patterns.

Comment 2. (1) Another key step of this work is that authors categorize animals into "grazing livestock" (e.g., buffalo, cattle, goats, horses, sheep) and "captive livestock" (e.g., chickens, ducks, pigs), and assume grazing species inhabit grasslands while captive species are confined to impervious surfaces (Lines 83-85). This simplification is problematic, as intensively raised livestock (e.g., pigs) frequently occupy peri-urban or rural agricultural lands rather than impermeable surfaces alone (https://doi.org/10.1016/j.oneear.2023.08.012). (2) In the reply letter, the authors cited Jiangsu pig farms to validate this classification, but subsequent checks revealed the mapped "impermeable surfaces" correspond to industrial calcium production facilities (name: 晶诚钙业 Jingcheng Calcium Industry 晶诚钙业 - 百度地图) rather than

pig farms. Given the global prevalence of industrial sites, such misclassifications risk severely compromising data product accuracy.

**Response:** (1) Thanks for the comment. In the revised manuscript, we have updated our suitability mask for all livestock to include agricultural lands, not only grassland and impervious surfaces. Thanks again for the recommended research, and we have now included this paper for better explanation. We have added a description of this improved method in Lines 128-129 of the paper and adjusted all the following result outputs, accuracy assessment and analysis accordingly (Figure 2-7). The model now better reflects real-world patterns, and the validation results also showed higher correlation coefficients as highlighted in the abstract. We believe this revision addresses the reviewer's concern, and we thank the reviewer for helping us improve the robustness of our approach.

(2) We appreciate the reviewer's careful check. We fully acknowledge the concern regarding the specific example cited in our previous reply. The coordinate we provided was based on a location labeled as a pig farm in a peer-reviewed publication focused on livestock facility detection in Jiangsu Province, as shown below:

Figure R1. Representative pig farm and industrial areas.

(The purple boundary denotes identified pig farms, while the red boundary indicates large-scale industrial facilities such as factories.)

This paper was published in 2020 (https://doi.org/10.19741/j.issn.1673-4831.2019.0764). We now understand that the current appearance of this location may correspond to a calcium industry facility. Nevertheless, we would like to emphasize that our study, much like the GLW (Gridded Livestock of the World) series developed by FAO and collaborators, does not aim to pinpoint the exact location of individual farms or facilities. Rather, our goal is to produce spatially and temporally continuous livestock density maps that reflect broader spatial patterns and temporal dynamics in livestock distribution at global scale. This distinction is crucial: the GLW products also rely on land use suitability and proxy variables (such as population density, land cover, and topography) rather than exact farm locations, due to the infeasibility of acquiring ground-truth farm locations globally, especially retrospectively over a multi-decade timespan. In this context, our methodology aligns with established livestock mapping practices. While individual mismatches (as the reviewer kindly pointed out) can occur, our model was validated against multiple GLW versions, along with county-level, city-level, provincial, and national statistics. The core objective of our suitability mask is to

guide probabilistic allocation of livestock within subnational units using ecological and socio-environmental proxies, not to directly geolocate farms.

Finally, we greatly appreciate the reviewer's feedback, which pushed us to reexamine and strengthen the core methodological framework of our study. Your input has played a critical role in enhancing the scientific quality and credibility of this work. Given the limited revision timeline, we have deposited an initial subset of the revised products (selected species/years together with the matching uncertainty layers) to our Zenodo repository (see Data Availability). We will continue to expand this record on a rolling basis to include the full set of maps and uncertainty layers, with clear versioning and a changelog to document updates. We believe that the revised manuscript and the new dataset will now meet your expectations.

---

## Author Response (AR3)

We would like to thank the editor and reviewers once more for the valuable comments and suggestions that greatly helped us to improve the manuscript. Thank you very much for your time and efforts.

Comment. I can see that the authors have made a reasonable effort to address critical comments from Reviewer #2, and their revisions are satisfactory. However, please ensure that uncertainty statements are included in both the abstract and conclusion, emphasizing the need for caution when considering these uncertainties in the use of this dataset.

**Response:** Thanks for the comment. We have now explicitly incorporated uncertainty statements in both the Abstract and the Conclusion, highlighting the need for caution of the uncertainties.

- (1) Abstract:
- "... We also release per-pixel, per-year uncertainty layers, and we recommend consulting these layers, especially for early decades and data-sparse regions."
  - (2) Conclusion:
- "... Additionally, we provide per-pixel, per-year uncertainty layers and advise cautious use, especially for early decades and data-sparse regions."
- (3) We also referenced the uncertainty layers in the Data Availability statement to ensure discoverability:
- "... Matching per-pixel, per-year uncertainty layers (0-1) are also provided; we recommend consulting them, especially for early decades and data-sparse regions."